# Safety and Outcomes of Inferior Vena Cava Filter Placement in Oncology Patients: A Single-Centre Experience

**DOI:** 10.3390/cancers16081562

**Published:** 2024-04-19

**Authors:** Paweł Kurzyna, Marta Banaszkiewicz, Michał Florczyk, Jarosław Kępski, Michał Piłka, Piotr Kędzierski, Rafał Mańczak, Piotr Szwed, Krzysztof Kasperowicz, Katarzyna Wrona, Grzegorz Doroszewski, Adam Torbicki, Marcin Kurzyna, Sebastian Szmit, Szymon Darocha

**Affiliations:** 1Chair and Department of Pulmonary Circulation, Thromboembolic Diseases and Cardiology, Centre of Postgraduate Medical Education, European Health Centre, ERN-LUNG Member, 05-400 Otwock, Poland; 2Department of Cardio-Oncology, Chair of Haematology and Transfusion Medicine, Centre of Postgraduate Medical Education, 01-813 Warsaw, Poland; 3Centre of Postgraduate Medical Education, Pelvic Injury and Pathology Department, Konarskiego 13, 05-400 Otwock, Poland

**Keywords:** inferior vena cava filter, venous thromboembolism, deep venous thrombosis, pulmonary embolism, cancer-associated thrombosis, oncology

## Abstract

**Simple Summary:**

In the oncology population, the risk of venous thromboembolism is significantly increased. Therefore, advanced therapeutic options, such as inferior vena cava (IVC) filters, can be an important part of treatment. Our observational, retrospective study compared the outcomes of IVC filter placement in the oncology population consisting of 62 patients and a non-oncology control group of 117 patients in the years 2012–2023. In both groups, there were no complications during IVC filter implantation procedures. In the oncology group, there was no recurrence of pulmonary embolism or deep vein thrombosis after filter implantation. There was no significant difference in other clinical outcomes between the two subgroups. Our study emphasized that the IVC filter is an effective method when standard anticoagulation treatment is not available for oncology patients. The use of inferior vena cava filters in this challenging population is also as safe as in non-oncology patients.

**Abstract:**

The risk of venous thromboembolism (VTE) in the oncology population is significantly higher than in non-cancer patients. Inferior vena cava (IVC) filters may, therefore, be an important part of VTE treatment. In this study, we address the outcomes of placing IVC filters in the oncology population. This single-centre, observational, retrospective study included 62 patients with active malignancy and acute VTE who underwent an IVC filter implantation due to contraindications to anticoagulation during the period 2012–2023. The control group consisted of 117 trauma patients. In both groups, an urgent surgical procedure requiring temporary cessation of anticoagulation was the most noted reason for IVC filter placement—76% in the oncology group vs. 100% in the non-oncology group (*p* < 0.001). No complications were reported during the IVC filter implantation procedures. There was no recurrence of pulmonary embolism or deep venous thrombosis in the oncology group after filter implantation. The rate of successful filter explantation, median time to retrieval, and abnormal findings during retrieval were not significantly different between both subgroups (64.3% vs. 76.5%, *p* = 0.334; 77 days vs. 84 days, *p* = 0.764; 61.5% vs. 54.2%, *p* = 0.672; respectively). The study showed that IVC filter placement is a safe and effective method of preventing PE in cancer patients with contraindications to anticoagulation. The complication rate following IVC filter implantation in cancer patients is low and similar to that in non-oncology patients.

## 1. Introduction

Venous thromboembolism (VTE) includes both deep vein thrombosis (DVT) and pulmonary embolism (PE). In the oncology population, this is a common complication. Compared to patients without cancer, cancer patients have a two to six times higher risk of VTE [1,2]. Active cancer is responsible for almost 20% of all new VTE events that occur in this community setting [1].

Anticoagulation (AC) is the mainstay of VTE treatment. However, in the case of oncology patients, AC may be contraindicated due to bleeding caused by the underlying malignancy or cancer treatment [3,4]. Consequently, in the treatment of VTE, inferior vena cava (IVC) filters have become commonplace [5,6]. This is in line with the latest European Society of Cardiology/European Respiratory Society Guidelines for Diagnosis and Treatment of Acute Pulmonary Embolism. Both contraindications to anticoagulant treatment and recurrent PE, despite adequate anticoagulant treatment, are indications for placing an IVC filter [7].

Despite the specific guidelines for IVC filter implantation, the extent to which they are used in daily clinical practice remains unclear. It can be challenging to apply the available date clinically, given the experience of the centres with filter placement procedures, the variability of the population with an indication for this procedure, and the variability of filter types. Consequently, there is a lack of studies investigating the use of IVC filters in the oncology population. In this study, we sought to investigate the characteristics of oncologic patients who are qualified for IVC filter placement and as well as to investigate what indications for the procedure are prevalent in this population. It seems that this may be crucial in the treatment process of cancer-associated thrombosis on a daily basis. Moreover, we address the safety and outcomes of IVC filter placement in the oncology population at our institution.

## 2. Materials and Methods

### 2.1. Patient Selection and Data Collection

That is a single-centre, observational, retrospective study of patients with active malignancy who underwent IVC filter implantation in the cardiology department between January 2012 and February 2023. The hospital database was searched for all patients discharged with IVC filter placement. Active malignancy was defined as the current need for anti-cancer therapy, or at least being eligible for such therapy, within the previous 30 days. For each confirmed case, baseline characteristics were recorded, including patient demographics, cancer type, history of recurrent VTE, current VTE manifestation, indications for filter placement, and rate of VTE recurrence. Outcome measures specific to filter retrieval included rate of filter retrievals, time to filter retrieval, and rate of procedural complications.

### 2.2. Procedural Details

The IVC filter implantation procedure is performed under local anaesthesia. Vascular access is obtained via the femoral vein or internal jugular vein, depending on the model of filter used. The following filters were used: Option™ Elite, OPTEASE™, Günther Tulip^®^, and Celect™ Platinum, according to manufacturer recommendations. Irrespective of the access vessel, a right-sided approach was preferred due to the direct path to the IVC and the minimal tilt of the filter. A diagnostic catheter (pigtail or MP) was positioned in the IVC to perform venography showing (depicting) the position of the renal veins and to exclude thrombosis of IVC. The preferred location for filter placement is caudal to the renal veins. The standard protocol for the procedure includes an X-ray check of the filter position at 24 h post-implantation. The following steps of the IVC filter implantation procedure are shown in Figure 1.

### 2.3. Statistics

The IBM SPSS Statistics package version 27.0 (IBM, Sheffield, UK) was used in the statistical analysis. Categorical variables were presented as numbers and percentages, while continuous variables were presented as mean and standard deviation (SD) or median with interquartile range (IQR), depending on their distribution. The χ^2^ test assessed categorical variables, while the Mann–Whitney test assessed continuous variables to identify differences between groups. There was a χ^2^ test applied to categorical variables with more than two categories. Statistical significance was determined with a *p*-value below 0.05.

## 3. Results

### 3.1. Patient Characteristics

Two hundred and seven IVC filters were implanted between January 2012 and February 2023 in the Department of Pulmonary Circulation, Thromboembolism and Cardiology at the European Health Centre in Otwock. Figure 2 shows the exclusion criteria. Finally, 179 patients were included in the analyses. There were 62 patients with active malignancy and 117 patients with an orthopaedic injury, which constituted the non-oncology group. The flowchart for patient enrolment is shown in Figure 2. Oncology patients were most likely to have acute PE (46.8% vs. 22.2%, *p* = 0.001) and a prior history of recurrent VTE (30.7% vs. 12.0%, *p* = 0.004), whereas, in the non-oncology group, isolated deep vein thrombosis was a more frequent VTE manifestation as a result of trauma and immobilisation (65.8% vs. 50.6%, *p* = 0.046). In the whole study population, there were no patients diagnosed with thrombophilia. Baseline patient characteristics are summarized in Table 1.

In the oncology group, filters were most commonly placed in patients with underlying colorectal cancer (24.2%), renal cell carcinoma (14.5%), and gynaecological malignancies (11.3%) (Figure 3). Filters were most commonly implanted in patients with disseminated cancer (40.3%) and locally advanced cancer (38.7%) and less frequently implanted in the group of patients with locally limited cancer (17.8%) and lymphoma. In addition, filters were also implanted in two (3.2%) patients with lymphoma (one patient in stage I and one patient in stage II according to the Ann Arbor classification).

In the orthopaedic group, filters were most commonly implanted in patients with pelvic (53%) and hip injuries (30.8%).

### 3.2. Indications for Filter Placement

In the total population studied, the most common reason for IVC filter placement was a recent or upcoming procedure requiring temporary cessation of anticoagulation. Specifically, 76% of patients in the oncology group had such a temporary contraindication to anticoagulation. Other contraindications to anticoagulation in the oncology population included pericardial bleeding (4%), gastrointestinal bleeding (6%), urinary bleeding (8%), and peripheral bleeding from the sarcoma tumour of the lower limb (2%). Additional indications cited included the failure of low-molecular-weight heparin (LMWH) (2%) and extensive vascular lesions of the tongue in the course of oral cancer (2%). A summary of the reported reasons for filter implantation is shown in Table 2.

### 3.3. VTE Recurrence

In the total study population, a new episode of acute pulmonary embolism occurred in 1/179 cases (0.6%) in a patient who underwent pelvic surgery. In addition, 60 patients from the non-oncology group underwent compression ultrasound examination at the time of filter removal due to clinical symptoms—progression of thrombosis or new vascular lesions were confirmed in 8/117 cases (6.8%). In the oncology population, there were no overt clinical episodes of VTE recurrence.

### 3.4. Filter Retrieval and Complications

Of the 179 patients, filter explantation was attempted in 96 (53.6%) cases. Attempted explantation was significantly less frequent in patients with cancer than in those without cancer (21.5% vs. 69.2%, *p* < 0.001). The rate of successful filter explantation in the total population studied was 74% and did not differ significantly between the studied subgroups (the oncology group: 64.3% vs. the non-oncology group: 76.5%, *p* = 0.334). The percentage of filters retrieved in both subgroups is shown in Figure 4.

For the filters that were retrieved, the median time to retrieval was 77 (IQR 17–292) days for the oncology patients and 84 (IQR 49–146) days for the non-oncology group and did not differ significantly between the subgroups (*p* = 0.764). The overall complication rate in the total population studied during the retrieval procedure was 72% (69 of 96 retrieved filters). The complication rate did not differ significantly between the subgroups. The most common abnormal finding during the retrieval procedure in both subgroups was the occurrence of a thrombus at the top of the filter (15.4% in the oncology group vs. 39.8% in the non-oncology group, *p* = 0.124). Specifically, a thrombus at the top of the filter occurred in a patient with locally limited prostate cancer and locally advanced colorectal cancer. Hook apposition was present in a patient with locally advanced renal cell carcinoma and a locally limited gynaecological tumour, as well as in a patient with locally limited multiple neoplasms. Clotted filter occurred in a patient with locally limited colorectal cancer and locally limited gynaecological cancer. These findings and other outcomes are summarized in Table 3. Moreover, there were no significant differences in the number of filters retrieved according to the stage of the cancer or according to the type of cancer (Table 4). Complications are shown in Figure 5.

## 4. Discussion

This is one of the few European studies to show that placing IVC filters in the oncology population is both beneficial and safe. The main findings of this study are as follows: i. IVC filter implantation is an effective mechanical method for the prevention of PE episodes in oncology patients contraindicated to anticoagulation; ii. IVC filters are as safe to use in the oncology population as in the non-oncology population; iii. the rate of filter explantation in the oncology population is not as high as the rate in the non-oncology patients.

There is a significant lack of prospective evidence on the safety and outcomes of IVC filter placement in the oncology population. The PREPIC (The Prévention du Risqué d’EmboliePulmonaire par Interruption Cave) study evaluated the clinical benefits of placing an IVC filter in 400 patients with proximal DVT at risk of PE and remains the biggest prospective randomized clinical trial published to date [8]. However, active cancer was diagnosed in only 14% (56/400) of patients in this study. In the trial, patients were randomly assigned to receive an IVC filter (200 patients) or no filter (200 patients). All the patients received anticoagulation treatment with LMWH or unfractionated heparin. At day 12, only 1.1% (two patients) of patients in the filter group and 4.8% (nine patients) of patients in the no-filter group were diagnosed with a symptomatic or asymptomatic PE event (odds ratio 0.22; 95% CI, 0.05–0.90). However, after two years, 20.8% of patients in the filter group and 11.6% of patients from the no-filter group had a recurrence of DVT (OR 1.87; 95% CI, 1.10–3.20). The authors’ conclusion was that there is an equivalent trade-off between fewer PE and more DVT with IVC filter placement. In particular, the oncology patients were at a higher risk of both recurrent VTE (HR 2.46) and death (HR 2.08) [8]. This conclusion was confirmed several years later by the results of the PREPIC 2 study [9]. In the study, patients after IVC filter implantation had a significantly lower incidence of symptomatic PE (6.2% vs. 15.1%; HR 0.37, CI 0.17–0.79, *p* = 0.008). However, they showed a higher cumulative incidence of symptomatic DVT (35.7% vs. 27.5%; HR 1.52, CI 1.02–2.27, *p* = 0.042) [9]. In view of the above results, the lack of recurrence of VTE in the oncology population in our study should be interpreted with a degree of caution. Firstly, these observations may be due to the fact that we did not routinely evaluate the deep venous system in asymptomatic patients at any time after filter implantation. Interestingly, similar observations were made by Schunn et al. Their matched, case–control study included 32 cancer patients, half without and half with IVF filters. They found no differences in terms of VTE recurrence between the subgroups. One recurrent PE was observed in the filter group and one in the control group. In addition, there were also no thromboembolic-related deaths reported in the study population [10].

However, the inherent thrombotic nature of malignancy combined with the poorer overall survival of the oncology population remains indisputable [11]. On the one hand, Barginear et al. addressed the benefits of IVC filter implantation on overall survival in 206 consecutive oncology patients with VTE. In the study, patients were classified into three treatment arms: IVC filter-only (77 patients), anticoagulation-only (62 patients), or a combination of both IVC filter and anticoagulation (67 patients). The authors observed that the median overall survival was significantly higher in the anticoagulation-only group (13 months) in comparison to both groups treated with IVC filter placement only (2 months) or the group treated with a combination of anticoagulation and IVC filters (3.25 months; *p* < 0.0002). Moreover, patients treated with IVC filter placement had a 1.9 times higher risk of death compared to patients who were only anticoagulated (HR = 0.528; 95% CI: 0.374–0.745) [12]. On the other hand, Stein et al., in their study, revealed that elderly cancer patients had a lower in-hospital all-cause mortality with IVC filters than those who did not have IVC filters (7.4% vs. 11.2%, *p* < 0.0001, respectively; relative risk 0.67). Moreover, within patients after IVC filter placement, the all-cause mortality rate within 3 months was lower than in patients who did not receive IVC filters (15.1% vs. 17.4%, *p* < 0.0001, respectively; relative risk of 0.86) [13].

It is, therefore, challenging to perform the risk–benefit assessment of IVC filter implantation in patients with advanced cancer. Many studies published to date have questioned the need for IVC filter placement. The survival rate of patients with advanced-stage disease is low. Therefore, the prevention of PE may be of little clinical benefit. In our study, the filter was implanted equally often in patients with disseminated cancer and locally advanced cancer. It was least frequently implanted in patients with locally advanced cancer. Jarrett et al., in their study of 116 cancer patients, found that 46% (42 patients) of 91 patients (78% of the total) with stage IV disease died within six weeks of having an IVC filter placed. Only 14% (16 patients) were alive at one year [14]. They also observed a statistically significant difference in survival (*p* < 0.001) when comparing patients with stage IV disease with patients with stage I-III disease [14]. Consequently, Wallace et al. showed that patients with stage IV solid tumours had a 76% survival rate at one month. It was only 29% at one year following IVC filter implantation [15]. In addition, patients with solid tumours classified as a local disease (*n* = 15), locally advanced disease (*n* = 95), and metastatic disease (*n* = 153) had 30-day survival probabilities of 0.93, 0.87, and 0.76, respectively. Patients with metastatic disease were 3.7 times more likely to die (*p* = 0.013) compared to patients with local disease [15]. This is in line with results obtained by Mansour et al. In the study, they retrospectively analysed 107 cancer patients (>90% of them had advanced-stage cancer) who had an IVC filter implanted. The authors revealed that among 59 patients with advanced-stage cancer, the median survival was 1.31 months (0.92–2.20). Specifically, 67.8% of patients survived less than three months, and 39% of patients survived less than one month [16]. Similar observations were made a few years later by American researchers. The study performed by Shaikh et al. included 179 cancer patients with retrievable IVC filters and 207 patients with permanent filters. Survival after filter placement was short and varied between the filter groups. The median time from filter implantation to death was 3.2 months in the permanent filter group and 8.9 months in the retrievable filter group [17]. Altogether, these results suggest that the benefits of IVC filter placement in patients with end-stage cancer are not evident. Moreover, these benefits may not fully account for the associated medical risks and costs.

Chau et al. analysed the cost-effectiveness of implanting IVC filters to prevent pulmonary embolism. Their study included 24 patients with brain tumours [18]. The authors concluded that the use of filters was effective in reducing the rate of recurrent PE. However, it was not cost-effective. The study also suggested that placing an IVC filter was more cost-effective in patients with a longer life expectancy. However, it should be noted that there were important limitations of this study. These included the small population and the lack of comparison between patients undergoing IVC filter placement and those who did not receive the therapy [18]. It should be stressed, however, that these are purely hypothetical considerations. According to the latest ESC/ERS guidelines, IVC filter placement is presently used when anticoagulant therapy is contraindicated or has failed [7]. The standard of care is that if the risk of the planned intervention is low, a medically proven therapy should not be refused because of cost.

Additional attention should be paid to the process of filter retrieval and also to the potential unique complications related to filter use and/or attempted retrieval. Published data suggest that retrieval rates are low, ranging from 11 to 46% [19,20,21]. As suggested by Karp et al., it is crucial to implement an intensive follow-up strategy to monitor patients’ progress after IVC filter implantation and to regularly analyse the appropriateness of filter retrieval. The authors noted that the retrieval rates of up to 60% have been reported with this approach. However, in the population of cancer patients, the filter removal rate was only 40%, despite using the above-mentioned approach [22]. Specifically, in the study conducted by Shaikh et al., only 40% of the 179 retrievable filters in oncology patients were retrieved [17]. This is in line with our observations, where filter removal was attempted in 21% of the oncology patients and as many as 71% of the control patients. Interestingly, the attempt to remove the filter was equally effective in both groups evaluated in our study (69% in the oncology population vs. 76% in the control group, *p* = 0.731). However, as we know from the study by Eifler et al., cancer is considered one of the risk factors for IVC filter retrieval failure (OR 3.27; 95% CI, 1.77–6.03) [23]. Other factors associated with filter permanency were age (OR, 1.03; 95% CI, 1.01–1.05), male sex (OR, 3.01; 95% CI, 1.64–5.54), and the failure of anticoagulation as an indication for filter implantation (OR, 8.12; 95% CI, 1.83–36.0) [23]. It is thought that the failure to remove the filter and convert the retrievable filter to a permanent one may be associated with an additional risk of complications for patients. A review of the MAUDE (Manufacturer and User Facility Device Experience) database, conducted by Andreoli et al., showed that between 2009 and 2012, as many as 86.6% of all 1606 complications recorded were related to retrievable filters, while only 13.2% were associated with permanent filters [24]. It should be noted that all filters used in our study were retrievable filters only, and we did not observe a high rate of associated complications.

IVC thrombosis is a well-documented complication of filter placement. According to available data, this complication occurs in 1.5% to 11% of cases, depending on the population and the type of filter used [25,26,27,28,29]. In the series by Athanasoulis et al., IVC thrombosis occurred in 4.4% (40 of 915) of patients diagnosed with cancer, compared with only 1.8% (15 of 816) of patients without cancer who received an IVC filter [29]. Given that cancer patients have a higher risk of recurrent VTE compared to non-oncology patients, the fact that caval thrombosis is more common in this population may be predictable [11,30,31]. Therefore, the most common complication recorded in our series was a clotted filter. Interestingly, this complication was similar in the cancer and non-cancer groups (23.1% of filters removed in the oncology group vs. 20.5% of filters removed in the non-oncology group, *p* = 1000). This is higher than the rates found in the study by Shaikh et al., where clotted filters occurred in 6.9% (5/72) of filters removed [17]. However, in our study, we did not record a single case of caval thrombosis. We hypothesize that this may be due to only a temporary contraindication to anticoagulation (surgery and short perioperative period) as the main indication for filter placement. Then, most patients were adequately anticoagulated most of the time, which prevented them from developing extended inferior vena cava thrombosis.

## 5. Limitations

Our study has several limitations, the most important being its retrospective, noncontrolled, single-centre nature. This may not only result in observational bias, but it is also the cause of incomplete observations and the lack of further cancer therapy and survival analysis or lack of a comparative control group of cancer patients who did not have an IVC filter implanted. Moreover, the single-centre nature of the study does not take into account whether other medical institutions have the appropriate conditions (including technical conditions but also operator experience and required equipment) to perform filter implantation procedures. It is, therefore, difficult to generalise our results and draw broad conclusions based on them.

## 6. Conclusions

In conclusion, IVC filter placement is a safe and highly effective method of preventing PE in selected cancer patients with VTE and contraindication to anticoagulation. Filter implantation in oncology patients is not associated with a higher risk of complications compared to the group of non-oncology patients. However, the most important element is proper qualification for the procedure in accordance with current guidelines. The lower probability of filter retrieval in oncology patients should also be considered in the qualification process. Further prospective, multicentre, randomized clinical trials are needed to better investigate the role of IVC filter implantation in the population of patients suffering from cancer-associated VTE. Survival analysis, including the analysis of pulmonary embolism-related mortality events, should be an important part of future studies.

## Figures and Tables

**Figure 1 cancers-16-01562-f001:**
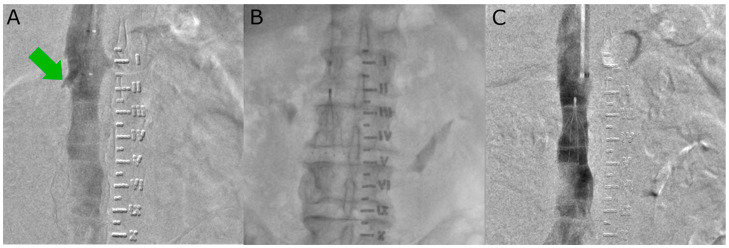
(**A**) Pre-implantation control cavography. The green arrow indicates the renal veins. (**B**) Moment of implantation of the filter in the inferior vena cava. (**C**) Post-implantation cavography to confirm proper position of the filter.

**Figure 2 cancers-16-01562-f002:**
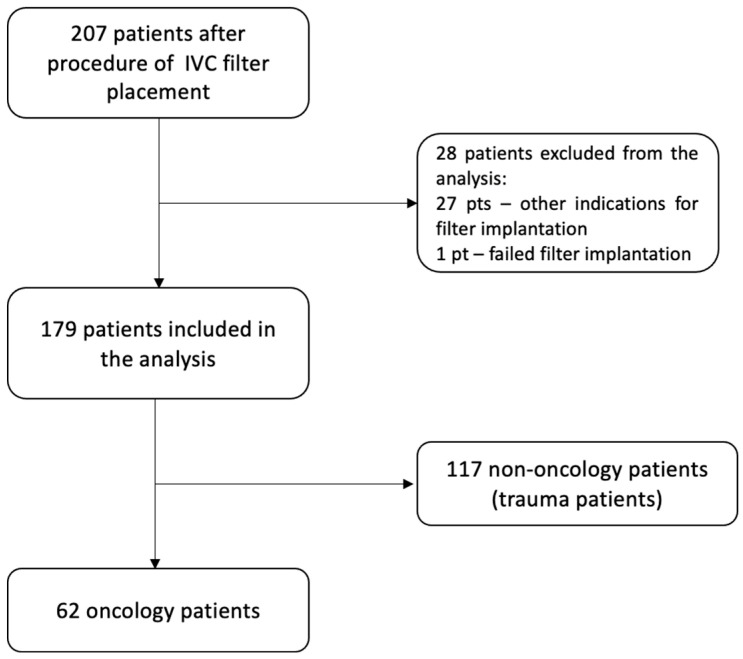
The flowchart showing the patient enrolment process.

**Figure 3 cancers-16-01562-f003:**
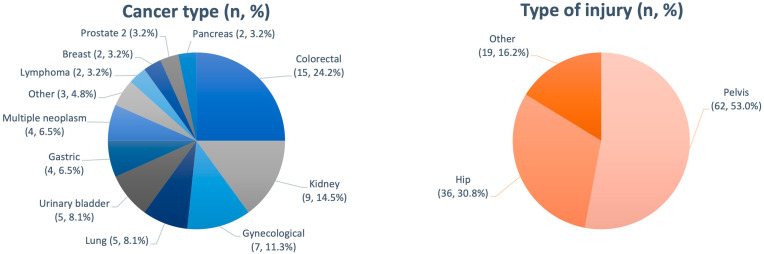
Number of patients after filter placement categorised by cancer type (oncology group) and by type of injury (non-oncology group).

**Figure 4 cancers-16-01562-f004:**
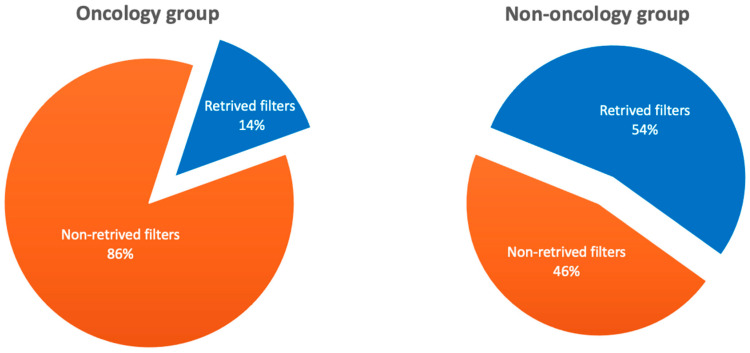
Percentage of filters removed in the oncology and non-oncology patient populations.

**Figure 5 cancers-16-01562-f005:**
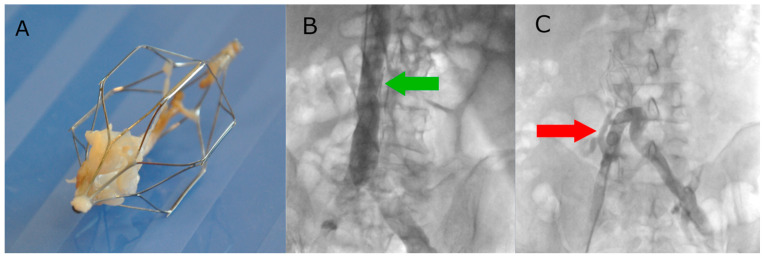
Filter retrieval complications: (**A**) macroscopic thrombus on the top of filter after explantation; (**B**) apposition of the hook to the vessel wall (green arrow); (**C**) clotted filter with marked collateral circulation (red arrow).

**Table 1 cancers-16-01562-t001:** Patient characteristic. VTE: venous thromboembolic disease.

Characteristic	All Patients (*n* = 179)	Oncology Patients (*n* = 62)	Non-Oncology Group (*n* = 117)	*p* Value
Median age (years) [mean ± SD]	60.1 ± 15.2	67.8 ± 11.2	56.0 ± 15.4	0.13
Male [*n*, %]	121 (67.6%)	37 (59.7%)	84 (71.8%)	<0.001
Recurrent VTE [*n*, %]	33 (18.4%)	19(30.7%)	14 (12.0%)	0.0004
**Manifestation of VTE**				
DVT [*n*, %]	66 (36.9%)	15 (24.2%)	51 (43.6%)	0.031
PE [*n*, %]	49 (27.4%)	22 (35.5%)	27 (23.1%)	0.090
DVT and PE [*n*, %]	64(35.8%)	25 (40.3%)	39 (33.3%)	0.361
**Filter model**				
Option™ Elite	75 (41.8%)	28 (45.1%)	47 (40.2%)	0.319
OPTEASE™	49 (27.4%)	12 (19.4%)	37 (31.6%)	
Günther Tulip^®^	20 (11.2%)	9 (14.5%)	11 (9.4%)	
Celect™ Platinum	35 (19.6%)	13 (21.0%)	22 (18.8%)	

**Table 2 cancers-16-01562-t002:** Indications for IVC filter placement.

Reason	Oncology Group	Non-Oncology Group	*p* Value
Recent/upcoming urgent procedure	47 (75.8%)	117 (100%)	<0.001
**Bleeding:**			
Pericardial	3 (4.8%)	-	
Gastrointestinal	4 (6.5%)	-	
Urinary	5 (8.1%)	-	
Other source	1 (1.6%)	-	
Failure of LMWH	1 (1.6%)	-	
Other	1 (1.6%)	-	

**Table 3 cancers-16-01562-t003:** Filter retrieval rates and complications.

	Oncology Group	Non-Oncology Group	*p*
Retrieval attempts [*n*, %]	13/62 (21%)	83/117 (71%)	<0.001
Filters retrieved [*n*, %]	9/13 (69%)	63/83 (76%)	0.731
Time to retrieval (days) [median, IQR]	77 (17–292)	84 (49–146)	0.764
**Abnormal findings during retrieval:**			
Thrombus at the top of the filter [*n*, %]	2/13 (15.4%)	33/83 (39.8%)	0.124
Apposition of the hook to the vessel wall [*n*, %]	3/13 (23.1%)	11/83 (13.3%)	0.397
Clotted filter [*n*, %]	3/13 (23.1%)	17/83 (20.5%)	1.000

**Table 4 cancers-16-01562-t004:** Filters retrieval rate depending on the stage of cancer and type of cancer.

	Filters Retrieved (*n*)	Filters Non-Retrieved (*n*)	*p* Value
**Cancer stage**			
Locally limited cancer	1	10	0.733
Locally advanced cancer	3	21	
Disseminated cancer	5	20	
Lymphoma	0	2	
**Localization of the neoplasm**			
Colorectal	3	12	
Kidney	2	7	
Gynecological	1	6	
Lung	0	5	
Urinary bladder	1	4	
Gastric	0	4	
Multiple neoplasm	1	3	
Sarcoma	0	2	
Lymphoma	0	2	
Breast	0	2	
Prostate	1	1	
Pancreas	0	2	
Other	0	3	

## Data Availability

The data presented in this study are no available due to ethical restrictions.

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
