# Peer review of "Safety and Outcomes of Inferior Vena Cava Filter Placement in Oncology Patients: A Single-Centre Experience"

_cancers, 2024, doi:10.3390/cancers16081562_

Round 1
Reviewer 1 Report
Comments and Suggestions for Authors
1. It has been explained in the introduction that there is currently a lack of research on the use of IVC filters in the tumor population, but the potential impact of the results of this study on clinical practice has not been introduced.
2. Does the reason for placing filters in tumor patients also include the common cause of thrombocytopenia caused by bone marrow suppression after chemotherapy.
3. It is recommended to statistically analyze the proportion of patients with acute lower limb DVT in the tumor group and non tumor group, as well as the proportion of patients with concomitant thrombophilia.
4. In the analysis of filter retrievals and compilations in the tumor group, subgroup analysis was not conducted for different types of tumors and different stages of tumors.
5. To ensure consistency in the style and format of the chart, it may be considered to modify the bar chart in Figure 3 to a pie chart in Figure 4, and also plot the disease classification of non tumor group patients in Figure 3.
6. This study analyzed the limitations, mainly retrospective, single center, and with a relatively small number of enrolled cases. There was a lack of comparison between patients who received IVC filter placement and those who did not receive treatment. Sharing these insights is very valuable for researchers attempting to replicate this study. In addition, limitations should also include the failure to consider whether different medical institutions have the conditions to perform IVC filter placement surgery (including changes in technology, operator experience, or equipment type).
7. In the discussion, future research areas were not proposed based on limitations, such as prospective, multicenter, and a control group without IVC filter placement was designed for tumors. The survival time of the experimental group and the control group was statistically analyzed, as well as the statistics of pulmonary embolism related mortality events.
Author Response
April 12th, 2024
Prof. Dr. Sebastiano Mercadante
Editor-in-Chief, Cancers
Dear Prof. Dr. Sebastiano Mercadante
Editor-in-Chief, Cancers,
Together with my co-authors, I would like to thank the Reviewers and Editorial Board of the Cancers for the review of the manuscript “Safety and outcomes of inferior vena cava filter placement in oncology patients: a single–centre experience”.
Following the suggestions of the Reviewers, the corrections have been made and marked in the current version of the manuscript. Responses to reviewers are provided below. All authors have approved the manuscript and agree with the current version. We re-submit the manuscript for consideration to be published.
Yours sincerely,
Marta Banaszkiewicz, MD, PhD
Department of Pulmonary Circulation, Thromboembolic Diseases and Cardiology
Center of Postgraduate Medical Education
European Health Center Otwock, Poland
Paweł Kurzyna, MD
Department of Pulmonary Circulation, Thromboembolic Diseases and Cardiology
Center of Postgraduate Medical Education
European Health Center Otwock, Poland
Reviewer 1
- It has been explained in the introduction that there is currently a lack of research on the use of IVC filters in the tumor population, but the potential impact of the results of this study on clinical practice has not been introduced.
Authors: Thank You for this comment. The “Introduction” section has been modified in accordance with your suggestion.
…
Despite the specific guidelines for IVC filter implantation, the extent to which they are used in daily clinical practice remains unclear. It can be challenging to apply the available date clinically, given the experience of the centres with filter placement procedures, the variability of the population with an indication for this procedure, and the variability of filter types. Consequently, there is a lack of studies investigating the use of IVC filter in the oncology population. In this study we sought to investigate the characteristics of oncologic patients who are qualified for IVC filter placement and as well as to investigate what indications for the procedure are prevalent in this population. It seems that this may be crucial in the treatment process of cancer-associated thrombosis on a daily basis. Moreover, we address safety and outcomes of IVC filter placement in the oncology population at our institution.
…
- Does the reason for placing filters in tumor patients also include the common cause of thrombocytopenia caused by bone marrow suppression after chemotherapy.
Authors: Indeed, thrombocytopenia as a contraindication to anticoagulation may be an indication for filter implantation in patients after chemotherapy. However, in our population there were no patients characterized by such an indication for the procedure. The indications for filter implantation in the study population are summarized in Table 2 in the 'results' section.
- It is recommended to statistically analyze the proportion of patients with acute lower limb DVT in the tumor group and non tumor group, as well as the proportion of patients with concomitant thrombophilia.
Authors: Thank You very much for this question. Analysis of the manifestation of VTE-including whether there was acute DVT, acute PE or acute PE + DVT- is included in the 'results' section. All data are included in Table 1. Within studied population, no patient was diagnosed with thrombophilia - this information has been added to the 'results' section.
Table 1. Patient characteristic. VTE: venous thromboembolic disease.
|
Characteristic |
All patients (n=179) |
Oncology patients (n=62) |
Non-oncology group (n=117) |
p value |
|
Median age (years) [mean ± SD] |
60.1 ± 15.2 |
67.8 ± 11.2 |
56.0 ± 15.4 |
0.13 |
|
Male [n, %] |
121 (67.6%) |
37 (59.7%) |
84 (71.8%) |
<0.001 |
|
Recurrent VTE [n, %] |
33 (18.4%) |
19(30.7%) |
14(12.0%) |
0.0004 |
|
Manifestation of VTE |
|
|
|
|
|
DVT [n, %] |
66 (36.9%) |
15 (24.2%) |
51 (43.6%) |
0.031 |
|
PE [n, %] |
49 (27.4%) |
22 (35.5%) |
27 (23.1%) |
0.090 |
|
DVT & PE [n, %] |
64(35.8%) |
25 (40.3%) |
39 (33.3%) |
0.361 |
|
Filter model |
|
|
|
|
|
Option™ Elite |
75 (41.8%) |
28 (45.1%) |
47 (40.2%) |
0.319 |
|
OPTEASE™ |
49 (27.4%) |
12 (19.4%) |
37 (31.6%) |
|
|
Günther Tulip® |
20 (11.2%) |
9 (14.5%) |
11 (9.4%) |
|
|
Celect™ Platinum |
35 (19.6%) |
13 (21.0%) |
22 (18.8%) |
Two hundred and seven IVC filters were implanted between January 2012 and February 2023 in the Department of Pulmonary Circulation, Thromboembolism and Cardiology at the European Health Centre in Otwock. Figure 2 shows the exclusion criteria. Finally, 179 patients were included in the analyses. There were 62 patients with active malignancy and 117 patients with an orthopaedic injury, which constituted the non-oncology group. The flowchart for patient enrolment is shown in Figure 2. Oncology patients were most likely to have acute PE (46.8% vs. 22.2%, p = 0.001) and a prior history of recurrent VTE (30.7% vs 12.0%, p = 0.004), whereas in the non-oncology group isolated deep vein thrombosis was a more frequent VTE manifestation, as a result of trauma and immobilisation (65.8% vs 50.6%, p = 0.046). In whole study population there were no patients diagnosed with thrombophilia. Baseline patient characteristics are summarised in Table 1.
- In the analysis of filter retrievals and compilations in the tumor group, subgroup analysis was not conducted for different types of tumors and different stages of tumors.
Authors: The results were supplemented with an analysis of the filter's retrievals and complications depending on the type of cancer and cancer stage.
Table 4. Filters retrieval rate depending on the stage of cancer and type of cancer.
|
|
Filters retrieved (n) |
Filters non-retrieved (n) |
p value |
|||
|
CCancer stage |
||||||
|
Disseminated cancer |
1 |
10 |
0.733 |
|||
|
Locally advanced cancer |
3 |
21 |
||||
|
Locally limited cancer |
5 |
20 |
||||
|
Lymphoma |
0 |
2 |
||||
|
Localization of the neoplasm |
||||||
|
Colorectal |
3 |
12 |
|
|||
|
Kidney |
2 |
7 |
||||
|
Gynecological |
1 |
6 |
||||
|
Lung |
0 |
5 |
||||
|
Urinary bladder |
1 |
4 |
||||
|
Gastric |
0 |
4 |
||||
|
Multiple neoplasm |
1 |
3 |
||||
|
Sarcoma |
0 |
2 |
||||
|
Lymphoma |
0 |
2 |
||||
|
Breast |
0 |
2 |
||||
|
Prostate |
1 |
1 |
||||
|
Pancreas |
0 |
2 |
||||
|
Other |
0 |
3 |
||||
…
For the filters that were retrieved, the median time to retrieval was 77 (IQR 17-292) days for the oncology patients and 84 (IQR 49-146) days for the non-oncology group and did not differ significantly between the subgroups (p=0.764). The overall complication rate in the total population studied during retrieval procedure was 72% (69 of 96 retrieved filters). The complication rate did not differ significantly between the subgroups. The most common abnormal finding during retrieval procedure in both subgroups was the occurrence of a thrombus at the top of the filter (15.4% in the oncology group vs. 39.8% in the non-oncology group, p=0.124). Specifically, thrombus at the top of the filter occurred in a patient with locally limited prostate cancer and locally advanced colorectal cancer. Hook apposition presented in a patient with locally advanced renal cell carcinoma, locally limited gynecological tumour and in a patient with locally limited multiple neoplasms. Clotted filter occurred in a patient with locally limited colorectal cancer and locally limited gynecological cancer. These findings and other outcomes are summarized in Table 3. Moreover, there were no significant differences in the number of filters retrieved according to the stage of the cancer or according to the type of cancer (Table 4). Complications are shown in Figure 5.
…
- To ensure consistency in the style and format of the chart, it may be considered to modify the bar chart in Figure 3 to a pie chart in Figure 4, and also plot the disease classification of non tumor group patients in Figure 3.
Authors: Thank You for this suggestion. The bar chart has been modified to a pie chart according to your suggestion. In addition, we have included the types of injuries in Figure 3.
Figure 3. Number of patients after filter placement categorised by cancer type(oncology group) and by type of the injury (non-oncology group.
- This study analyzed the limitations, mainly retrospective, single center, and with a relatively small number of enrolled cases. There was a lack of comparison between patients who received IVC filter placement and those who did not receive treatment. Sharing these insights is very valuable for researchers attempting to replicate this study. In addition, limitations should also include the failure to consider whether different medical institutions have the conditions to perform IVC filter placement surgery (including changes in technology, operator experience, or equipment type).
Authors: Thank you for this remark. The 'limitations' section has been modified in accordance with your suggestion.
5. Limitations
Our study has several limitations, and the most important being its retrospective, noncontrolled, single-centre nature. This may not only result in observational bias, but it is also the cause of incomplete observations - lack of further cancer therapy and survival analysis, or lack of a comparative control group of cancer patients who did not have an IVC filter implanted. Moreover, the single-center nature of the study does not take into account whether other medical institutions have the appropriate conditions (including technical conditions, but also operator experience and required equipment) to perform filter implantation procedures. It is therefore difficult to generalise our results and draw broad conclusions based on them.
- In the discussion, future research areas were not proposed based on limitations, such as prospective, multicenter, and a control group without IVC filter placement was designed for tumors. The survival time of the experimental group and the control group was statistically analyzed, as well as the statistics of pulmonary embolism related mortality events.
Authors: Thank You for this remark. The 'Conclusions’' section has been modified in accordance with your suggestion.
6. Conclusions
In conclusion, IVC filter placement is a safe and highly effective method of preventing PE in selected cancer patients with VTE and contraindication to anticoagulation. Filter implantation in oncology patients is not associated with a higher risk of complications compared to the group of non-oncology patients. However, the most important element is proper qualification for the procedure, in accordance with current guidelines. The lower probability of filter retrieval in the oncology patients should also be considered in the qualification process. Further prospective, multicenter, randomized clinical trials are needed to better investigate the role of IVC filter implantation in the population of patients suffered from cancer-associated VTE. Survival analysis, including analysis of pulmonary embolism related mortality events, should be an important part of future studies.

Reviewer 2 Report
Comments and Suggestions for Authors
A RBI (Helsinky committee) is mandatory !!! Please confirm that you have an agreement.
Was there a difference between various types of filters?
It is necessary to introduce references in lines 179-182, 192-193.
I think it is necessary to give us in "references" all the studies on the same subject ("This is one of the few European studies demonstrating the benefits and safety of IVC 171 filter placement among the oncology population") and discuss the results.
What is new in your study? " Filter implantation in oncology patients is not associated with a higher risk of complications compared to the group of non-oncology patients". If the same results are found in other studies, you bring nothing new. In this case, it will be quite difficult to recommend the publication of your study.
About the references: this journal accepts 10-11 authors followed by "et al"?
The Vancouver "style" is different.
Author Response
Reviewer 2
- A RBI (Helsinky committee) is mandatory !!! Please confirm that you have an agreement.
Authors: Thank You for this remark. Thank you for this comment. However, according to the current law in our country, the approval of the bioethics committee is not required for observational retrospective studies, and this is in accordance with the Declaration of Helsinki.
- Was there a difference between various types of filters?
Authors: Thank You very much for this question. There was no statistically significant difference in the use of each type of filter in both oncology and non-oncology patient populations. Data on the types of filters implanted in both subpopulations and in the entire study population have been added to Table 1.
Table 1. Patient characteristic. VTE: venous thromboembolic disease.
|
Characteristic |
All patients (n=179) |
Oncology patients (n=62) |
Non-oncology group (n=117) |
p value |
|
Median age (years) [mean ± SD] |
60.1 ± 15.2 |
67.8 ± 11.2 |
56.0 ± 15.4 |
0.13 |
|
Male [n, %] |
121 (67.6%) |
37 (59.7%) |
84 (71.8%) |
<0.001 |
|
Recurrent VTE [n, %] |
33 (18.4%) |
19(30.7%) |
14(12.0%) |
0.0004 |
|
Manifestation of VTE |
|
|
|
|
|
DVT [n, %] |
66 (36.9%) |
15 (24.2%) |
51 (43.6%) |
0.031 |
|
PE [n, %] |
49 (27.4%) |
22 (35.5%) |
27 (23.1%) |
0.090 |
|
DVT & PE [n, %] |
64(35.8%) |
25 (40.3%) |
39 (33.3%) |
0.361 |
|
Filter model |
|
|
|
|
|
Option™ Elite |
75 (41.8%) |
28 (45.1%) |
47 (40.2%) |
0.319 |
|
OPTEASE™ |
49 (27.4%) |
12 (19.4%) |
37 (31.6%) |
|
|
Günther Tulip® |
20 (11.2%) |
9 (14.5%) |
11 (9.4%) |
|
|
Celect™ Platinum |
35 (19.6%) |
13 (21.0%) |
22 (18.8%) |
- It is necessary to introduce references in lines 179-182, 192-193.
Authors: Thank You for this remark. References have been added to the lines you provided.
- I think it is necessary to give us in "references" all the studies on the same subject ("This is one of the few European studies demonstrating the benefits and safety of IVC 171 filter placement among the oncology population")and discuss the results.
Authors: Thank You for Your remark. In accordance with your suggestion, we have added 3 new references, the results of which have been carefully analyzed and discussed by us. In our opinion, all noteworthy studies on this topic have been included in the references and discussed in the study.
…
On the one hand, Barginear et al addressed the benefits of IVC filters implantation on overall survival in 206 consecutive oncology patients with VTE. In the study patients were classified into three treatments arms: IVC filter – only (77 patients), anticoagulation-only (62 patients) or combination of both IVC-filter and anticoagulation (67 patients). Authors observed, that median overall survival was significantly higher in the group of anticoagulation-only (13 months) in comparison to the both group treated with IVC filter placement only (2 months) or group treated with combination of anticoagulation and IVC filters 3.25 months; p < 0.0002). Moreover, patients treated with IVC filters placement had 1.9 times higher risk of death compared to patients who were only anticoagulated (HR = 0.528; 95% CI: 0.374-0.745) [12]. On the other hand, Stein et al in their study revealed, that elderly cancers patients had a lower in-hospital all-cause mortality with IVC filters than those who did not have IVC filters (7.4% vs. 11.2%, p <0.0001, respectively; relative risk 0.67). Moreover, within patients after IVC filters placement all-cause mortality rate within 3 months was lower than in patients who did not receive IVC filters (15.1% vs 17.4%, p < 0.0001, respectively; relative risk 0.86) [13]
…
This is in line with results obtained by Mansour et al. In the study they retrospectively analyzed 107 cancer patients (>90% of them had with advanced-stage cancer), who had IVC filter implanted. The authors revealed that among 59 patients with advanced stage cancer the median survival was 1.31 months (0.92-2.20). Specifically, 67.8% of patients survived less than three months and 39% of patients survived less than one month [16].
- What is new in your study?" Filter implantation in oncology patients is not associated with a higher risk of complications compared to the group of non-oncology patients". If the same results are found in other studies, you bring nothing new. In this case, it will be quite difficult to recommend the publication of your study.
Authors: Thank You very much for this question. In the study we demonstrate that IVC filter placement is a safe and highly effective method of preventing PE in selected cancer patients with VTE and contraindication to anticoagulation. These conclusions are indeed consistent with the results of studies obtained to date by other researchers, which only makes them even more reliable. We are convinced that the results are an important voice coming from a reference center for the treatment of venous thromboembolism, in the ongoing discussion regarding the difficulties of anticoagulant treatment among oncology patients. Importantly, more than 60 consecutive patients were included in the study, in which not only the effectiveness and safety of the filter implantation procedure itself, but also the indications for the procedure were analyzed. It is the proper qualification for filter implantation that makes this procedure the most effective and safe, and this is what we wanted to emphasize in presenting our results. On the other hand, the rather low filter retrieval rate shows what still needs to be worked on and improved in the treatment of VTE in cancer patients. Therefore, we believe that the publication of the obtained results will be helpful primarily to clinicians, working on a daily basis with patients suffered from cancer-associated thrombosis.
- About the references: this journal accepts 10-11 authors followed by "et al"? The Vancouver "style" is different.
Authors: Thank You for this remark. Indeed, the references were formatted in 'MDPI' style, we corrected them so that they are now in 'Vancouver' style as required by the journal.
